# Evaluation of Chemical Composition and Anti-Staphylococcal Activity of Essential Oils from Leaves of Two Indigenous Plant Species, *Litsea leytensis* and *Piper philippinum*

**DOI:** 10.3390/plants13243555

**Published:** 2024-12-20

**Authors:** Genesis Albarico, Klara Urbanova, Marketa Houdkova, Marlito Bande, Edgardo Tulin, Tersia Kokoskova, Ladislav Kokoska

**Affiliations:** 1Department of Pure and Applied Chemistry, Visayas State University, Visca, Baybay City 6521-A, Leyte, Philippines; genesis.albarico@vsu.edu.ph; 2Department of Crop Science and Agroforestry, Faculty of Tropical AgriSciences, Czech University of Life Sciences Prague, Kamycka 129, 165 00 Prague, Czech Republic; houdkovam@ftz.czu.cz; 3Department of Sustainable Technologies, Faculty of Tropical AgriSciences, Czech University of Life Sciences Prague, Kamycka 129, 165 00 Prague, Czech Republic; urbanovak@ftz.czu.cz; 4Institute of Tropical Ecology, Visayas State University, Visca, Baybay City 6521-A, Leyte, Philippines; marlitojose.bande@vsu.edu.ph; 5Philrootcrops, Visayas State University, Baybay City 6521-A, Leyte, Philippines; edgardo.tulin@vsu.edu; 6Department of Animal Science and Food Processing, Faculty of Tropical AgriSciences, Czech University of Life Sciences Prague, Kamycka 129, 165 00 Prague, Czech Republic; kokoskova@ftz.czu.cz

**Keywords:** essential oil, GC-MS, hydrodistillation, Lauraceae, Piperaceae, volatile compounds

## Abstract

Many indigenous plants of the Philippines, including essential oil-bearing species, remain phytochemically and pharmacologically unexplored. In this study, the chemical composition of leaf essential oils (EOs) hydrodistilled from *Litsea leytensis* (Lauraceae) and *Piper philippinum* (Piperaceae) was determined using dual-column (HP-5MS/DB-WAX)/dual-detector gas chromatography and mass spectrometry analysis. Caryophyllene oxide (15.751/16.018%) was identified as the main compound in *L. leytensis* EO, followed by β-caryophyllene (11.130/11.430%) and α-copaene (9.039/9.221%). Ishwarane (25.937/25.280%), nerolidol (9.372/10.519%) and 3-ishwarone (6.916/2.588%) were the most abundant constituents of *P. philippinum* EO. Additionally, the in vitro growth-inhibitory activity of the EOs in the liquid and vapour phases against *Staphylococcus aureus* was evaluated using the broth microdilution volatilisation assay. Although the results showed no anti-staphylococcal effect, the presence of various bioactive compounds in both EOs suggests their potential future use in industrial applications.

## 1. Introduction

Indigenous plants, especially endemic species, are less explored as sources of new phytochemicals, including those found in essential oils (EOs), with potential industrial uses [1,2]. Most of these plants are found in specific regions known as biodiversity hotspots. The Philippines, a tropical archipelago located in Southeast Asia, is considered one of the world’s best biodiversity hotspots, with more than 5800 species of endemic plants, including EO-bearing flora [3]. Various EOs isolated from the numerous native plants of the Philippines have been recognised worldwide for their commercial and economic importance, and many of them are currently utilised in the cosmeceutical and pharmaceutical industries. For example, EO distilled from the resin of *Canarium luzonicum*, commonly known as Manila elemi, is used as a fragrance in soap and perfumes and as a base for liniments [4]. Moreover, other economically valuable EOs can also be obtained from plant species commonly belonging to the Annonaceae, Burseraceae, Lauraceae, Piperaceae and Zingiberaceae families, among others. However, the EO chemical composition of some species belonging to these families has not yet been analysed. One typical example is *Litsea leytensis* Merr. (Lauraceae), a species endemic in the Philippines and commonly known as batikuling or bitokling. It is a medium-sized tree that inhabits low- and medium-altitude forests in Luzon and the Eastern Visayas regions [5]. Light-to-medium-weight wood from the tree is used locally for pattern making, ceilings and carving due to its scent that naturally repels termites, ants and woodworms [6]. Another example of a native Philippine plant is *Piper philippinum* Miq. (Piperaceae), which is a woody climber distributed throughout southern China, Taiwan and the Philippines in thickets and forests at low and medium altitudes [7]. The older stems are traditionally used for betel quid chewing, together with lime and catechu. With the exception of bioactive lignans isolated from *P. philippinum* [8], there is currently no information about the chemistry and biological activity of either of these species. Since certain species of the genera *Litsea* and *Piper* (e.g., *Litsea cubeba* and *Piper nigrum*) are important industrial sources of EOs, we decided to determine the chemical composition of the volatile compounds present in the leaves of two indigenous Philippine plants, namely *L. leytensis* and *P. philippinum*, using gas chromatography and mass spectrometry (GC-MS). Additionally, in vitro anti-staphylococcal susceptibility testing of these two EOs was performed in both the liquid and vapour phases.

## 2. Results and Discussion

Hydrodistillation of the leaves *L. leytensis* and *P. philippinum* produced light, yellow-coloured EOs with yields (*v*/*w*) of 0.14% and 0.77% on a dry plant weight basis, respectively. According to the recommendations previously proposed for more effective assessment of the anti-infective potential of natural products, minimum inhibitory concentrations (MICs) below 100 μg/mL for EOs should be considered as an indicator of promising activity. In addition, samples with MICs higher than 1000 μg/mL should strictly be evaluated as not active and excluded from further experiments [9]. Based on these criteria, the EOs did not show any antibacterial activity against *S. aureus* in the liquid nor vapour phases (MICs > 1024 μg/mL). The complete results of the chemical analysis and the composition of *L. leytensis* and *P. philippinum* EOs are provided in Table 1 and Table 2.

Based on the GC-MS analysis using HP-5MS/DB-WAX columns, a total of 68/61 and 54/48 compounds were identified in the samples of *L. leytensis* and *P. philippinum*, representing 93.775/90.128% and 88.524/89.045% of the total content, respectively. Analysis revealed that monoterpenes and sesquiterpenes were the predominant chemical classes within the major constituents of the tested EOs. For *L. leytensis*, 1.825/1.387% monoterpenes and 87.319/81.585% sesquiterpenes were identified, respectively, using HP-5MS/DB-WAX columns, while 7.245/10.807% monoterpenes and 80.384/73.709% sesquiterpenes were identified for *P. philippinum*. Caryophyllene oxide (15.751/16.018%) was the main component of *L. leytensis* EO, followed by β-caryophyllene (11.130/11.430%) and α-copaene (9.039/9.221%). All three compounds are known to exhibit various biological activities. For example, analgesic, antibacterial, anticancer, antifungal, anti-inflammatory, antioxidant, cardioprotective, gastroprotective, hepatoprotective, immunomodulatory, nephroprotective and neuroprotective effects were reported for β-caryophyllene [12,13,14]. Likewise, caryophyllene oxide has analgesic and anti-inflammatory properties [15]. Besides its antioxidant and neuroprotective activities [16], α-copaene is a common attractant to insect pests such as the Mediterranean fruit fly [17] and redbay ambrosia beetles [18]. In previous experiments investigating their antimicrobial properties, caryophyllene oxide, β-caryophyllene and α-copaene produced a moderate-to-weak in vitro growth-inhibitory effect against *Staphylococcus aureus* with MICs of 125, 250 and ~1000 μg/mL, respectively [19,20]. The relatively high MIC values previously detected for all three compounds can explain the absence of anti-staphylococcal activity of EOs assayed in this study. According to the results of our analysis, the chemical composition of *L. leytensis* EO significantly differs from that of *L. cubeba*, which is the most important species of the genus economically. Its EO is used in the perfume industry as a commercial source of citral [21]. Although the chemistry of the EO from the leaf of *L. cubeba* varies significantly depending on the geographical origin of the sample, with 1,8-cineole or linalool being the main constituents [22], its chemical composition differs significantly from that of *L. leytensis* leaf EO. Previously published analyses of EOs from other species of the *Litsea* genus suggest that the chemical composition of *L. deccanensis* left EO is more similar to that of *L. leytensis*, as it also contains β-caryophyllene and caryophyllene oxide as the main components [22,23]. Reported chemical variability in EOs obtained from leaves of *L. cubeba* of different geographical origin [24] suggests that environmental factors can also affect the phytochemical variability of other species of the genus, including composition of *L. leytensis* EO. In addition, the experiment comparing solvent-free microwave extraction and hydrodistillation showed that the chemical composition of *L. cubeba* EO could be influenced by the extraction method used [25], which should be considered for further scaling-up of the extraction of volatile compounds from species of the *Litsea* genus.

The chemical analysis showed that ishwarane (25.937/25.280%) is a major component of *P. philippinum* leaf EO, followed by nerolidol (9.372/10.519%) and 3-ishwarone (6.916/2.588%). The chemical profile of *P. nigrum*, the most economically and industrially important species of the genus, differs therefore significantly from *P. philippinum* EO [26,27,28,29]. Nevertheless, leaf EOs of other *Piper* species, such as *P. arboretum*, *P. aduncum* and *P. guadianum*, also contain a significant amount of δ-cadinene, β-caryophyllene and nerolidol [28]. Nerolidol is an economically important sesquiterpene since it is predominantly utilised as a fragrance component in the perfume industry [30] with known anticancer, anti-inflammatory, antimicrobial, antinociceptive, antioxidant and antiparasitic (anti-leishmanial, -malarial, -schistosomal and -trypanosomal) activities [31,32,33,34]. On the other hand, ishwarane is a rare sesquiterpene among the species of the *Piper* genus [35], which is only found in the leaf EO of *P. fulvescens* [36] and *P. alatipetiolatum* [37], as well as in the fruit EO of *P. guineense* [38]. Ishwarane was reported to have antifungal activity against *Cladosporium cladosporioides* [35]. 3-Ishwarone, another rare sesquiterpene detected in *P. philippinum*, was also found in the leaf EO of *Peperomia oreophila* [39] and *Peperomia scandens* [40]. Since considerable variations in chemical composition of EOs have previously been demonstrated for different populations of *P. nigrum* grown in China [41], the potential influence of environmental conditions on the chemistry of plants belonging to the *Piper* genus should be considered prior to the industrial utilisation of their EOs. It has also recently been reported that the chemical composition of *P. nigrum* hydrodistilled EO varies from its supercritical carbon dioxide extract [42]. This result suggests that the analysis of composition of volatile compounds present in the *Piper* genus can be influenced by the extraction method used.

The separation of EOs using GC on stationary phases of different polarity provides an analytical tool useful for various applications, the most notable being the confirmation of specific isomers [43]. Effective separation of several isomers from the two plant EOs in the current study was conducted using non-polar HP-5MS and polar DB-WAX columns. Using this approach, isomeric cadinenes detected in *L. leytensis* EO were identified, whereas α- and γ-cadinene were found on HP-5MS, and δ-cadinene was found on DB-WAX only. In the case of *P. philippinum* EO, effective separation was achieved for α- and β-copaene on both columns, and their corresponding isomer, ylangene, was found only when using DB-WAX. Moreover, the polar DB-WAX column offered additional information regarding the components of each EO sample, such as the detection of hexadecenoic acid in *P. philippinum* EO.

## 3. Materials and Methods

### 3.1. Chemicals and Reagents

Camphene (CAS 79-92-5), β-caryophyllene (CAS 87-44-5), geraniol (CAS 106-24-1), α-caryophyllene (CAS 6753-98-6), linalool (CAS 126-91-0), methyl octanoate (CAS 111-11-5), myrcene (CAS 123-35-3), α-pinene (CAS 7785-70-8) and β-pinene (CAS 18172-67-3) were used as analytical standards. Furthermore, *n*-alkanes (ranging from C_8_ to C_40_) were used as calibration standards. With the exception of *n*-hexane (CAS 110-54-3; Merck, Darmstadt, Germany), which was used as a solvent for the preparation of analytical EO samples, all other chemicals were obtained from Sigma-Aldrich (Prague, Czech Republic).

### 3.2. Plant Material

The leaves of *L. leytensis* and the aerial parts of *P. philippinum* were collected during January 2019 at the base of Mount Pangasugan, Leyte Island, the Philippines. The plants were authenticated at the Jose Vera Santos Memorial Herbarium of the College of Science of the University of the Philippines by plant taxonomist Edwino S. Fernando. Voucher specimens were deposited at CZU, in the herbarium of the Department of Botany and Plant Physiology, Faculty of Agrobiology, Food and Natural Resources: 02576KBFRB (*L. leytensis*) and 02576KBFRB (*P. philippinum*). For EO extraction, separate air-dried plant samples were homogenised (Grindomix GM 100, Retsch, Haan, Germany). Residual moisture content was analysed in triplicate using a Scaltec SMO 01 (Scaltec Instruments, Gottingen, Germany) at 130 °C.

### 3.3. Hydrodistillation

The EOs from *L. leytensis* and *P. philippinum* were extracted via hydrodistillation from air-dried plant material. A Clevenger-type apparatus (Merci, Brno, Czech Republic) was used to extract the EO from the material placed in 1 L of distilled water for 3 h, according to the European Pharmacopoeia [44]. After distillation, the EOs were stored in sealed glass vials at 4 °C until analysed.

### 3.4. Bacterial Strain and Culture Media

*S. aureus* standard strain ATCC 29213 was cultivated in Mueller–Hinton broth and agar, both purchased from Oxoid (Basingstoke, UK). The pH of the broth was equilibrated to 7.6 with Trizma base (Sigma-Aldrich).

A stock culture of *S. aureus* was cultivated at 37 °C for 24 h prior to susceptibility testing. The turbidity of the bacterial suspension was then adjusted to 0.5 McFarland standard, using a Densi-La-Meter II (Lachema, Brno, Czech Republic), to obtain a final concentration of 10^7^ CFU/mL. Susceptibility of the bacterium to oxacillin (86.3%, CAS 7240-38-2; Sigma-Aldrich) was utilised as a positive antibiotic control [45].

### 3.5. Antimicrobial Assay

The antibacterial potential of the plant EOs was assessed, in both the liquid and vapour phase, using the broth microdilution volatilisation method [46]. Standard 96-well immune plates with flanged lids designed to reduce evaporation (SPL Life Sciences, Naechon-Myeon, Pocheon-si, Republic of Korea) were utilised. First, 30 μL of agar was pipetted into each flange, except the outermost flanges, and inoculated with 5 μL of the bacterial suspension. Secondly, samples of the EOs were dissolved in dimethylsulfoxide (DMSO) (Sigma-Aldrich) at a maximum concentration of 1% and diluted in broth medium. Serial dilutions were prepared from the samples of both EOs (seven two-fold dilutions), starting at 1024.00 μg/mL. A 96-pin multi-blot replicator (National Institute of Public Health, Prague, Czech Republic) was used to inoculate the plates with the bacterial suspension. Wells containing inoculated and non-inoculated broth were used as growth and purity controls simultaneously. Lastly, the plates and lids were fastened together to ensure an air-tight fit using clamps (Lux Tool, Prague, Czech Republic) and handmade wooden pads and incubated at 37 °C for 24 h. The MICs were evaluated by visual assessment. A metabolically active bacterial colony was coloured with thiazolyl blue tetrazolium bromide dye (Sigma-Aldrich) at a concentration of 600.00 μg/mL. When the colour changed from yellow to purple (relative to the colours in the control wells and flanges), the endpoint (MIC value) was recorded in the broth and agar. The MIC values (μg/mL) were the lowest concentrations capable of inhibiting bacterial growth, compared with the compound-free control. The negative control containing 1% of DMSO did not inhibit the growth of the strain tested, neither in the broth or agar media. All experiments were performed in triplicate, in three independent experiments. According to the widely accepted norm in MIC testing, the mode and median were used for the final value calculation when the triplicate endpoints were within the two- and three-dilution range, respectively [47].

### 3.6. GC-MS Analysis

GC-MS analysis was used to determine the main components of the EOs. An Agilent GC-7890B dual-column/dual-detector gas chromatograph (Agilent Technologies, Santa Clara, CA, USA) was utilised, equipped with an Agilent 7693 autosampler, two columns, a fused-silica HP-5MS column (30 m × 0.25 mm, film thickness 0.25 µm, Agilent 19091s-433) and a DB-WAX column (30 m × 0.25 mm, film thickness 0.25 µm, Agilent 122–7132), and a flame ionisation detector (FID) coupled with a single quadrupole mass selective detector Agilent MSD-5977B (Agilent Technologies, Santa Clara, CA, USA). Helium was used as a carrier gas (1 mL/min) and the injector temperature for both columns was set at 250 °C. The oven temperature was increased after 3 min from 50 to 280 °C for both columns. After an isothermic period of 3 min, a heating rate of 3 °C/min was used until 120 °C, after which 5 °C/min was utilised until 250 °C. This was followed by a 5 min holding time at 250 °C, after which the heating rate increased to 15 °C/min until 280 °C. An isothermic period of 20 min followed. EOs were diluted to a concentration of 20 µL/mL in *n*-hexane, and, subsequently, 1 µL of each sample was injected into the GC-MS in a split mode (split ratio 1:50). The mass detector conditions were as follows: ionisation energy 70 eV, ion source temperature 230 °C, scan time 1 s and mass range 40–600 m/z. Identification of the constituents was performed by comparing their retention indices (RIs), retention times (RTs) and spectra with those in the National Institute of Standards and Technology Library ver. 2.0.f (NIST, Gaithersburg, MD, USA) [10,11], as well as against authentic standards (Sigma-Aldrich) and with the literature. The RIs were calculated using the RTs of the *n*-alkane series (ranging from C_8_ to C_40_) for compounds separated on the HP5-MS column. The relative percentages of the EO components were determined on both columns using FID. The analysis of each EO was performed in three independent measurements and the relative peak area percentage was expressed as an average plus/minus standard deviation.

## 4. Conclusions

This is the first report of the chemical composition of EOs hydrodistilled from the leaves of *L. leytensis* (Lauraceae) and *P. philippinum* (Piperaceae), which provides new insights into the phytochemistry of these indigenous plant species found in the Philippines. Sesquiterpenoids, namely caryophyllene oxide, α-copaene and β-caryophyllene (*L. leytensis*), and 3-ishwarone, ishwarane and nerolidol (*P. philippinum*), were the predominant class of compounds identified in both EOs. Using GC-MS equipped with two columns of differing polarity, isomeric cadinenes were detected in *L. leytensis* EO and copaenes and ylangene were identified in *P. philippinum* EO. Although the assessment of the antibacterial activity of the EOs showed no growth-inhibitory effect on *S. aureus*, the presence of bioactive compounds such as caryophyllene oxide, β-caryophyllene, α-copaene, ishwarane and nerolidol in the EOs suggests their potential future use in cosmetic and pharmaceutical applications. However, the evaluation of safety and knowledge of a broader spectrum of biological properties, such as anticancer, antifungal and antioxidant effects, will be necessary before their possible industrial usage.

## Figures and Tables

**Table 1 plants-13-03555-t001:** *Litsea leytensis* leaf essential oil chemical composition.

^a^ RI		Compounds	^b^ Cl.	^c^ Content [%]		^d,e^ Identification
Obs.	Lit.			HP-5MS	DB-WAX	HP-5MS
929	939	α-Pinene	MH	0.234 ± 0.002	0.183 ± 0.035	RI, MS, Std.
944	943	Camphene	MH	0.022 ± 0.002	-	RI, MS, Std
972	980	β-Pinene	MH	0.090 ± 0.001	0.074 ± 0.011	RI, MS, Std
990	989	2-Amylfuran	O	0.037 ± 0.000	0.022 ± 0.002	RI, MS
1022	1026	β-Cymene	MH	0.090 ± 0.003	0.077 ± 0.024	RI, MS
1026	1031	D-Limonene	MH	0.057 ± 0.001	0.043 ± 0.006	RI, MS
1028	1033	Eucalyptol	OM	0.247 ± 0.001	0.195 ± 0.014	RI, MS
1099	1098	Linalool	OM	0.120 ± 0.006	0.059 ± 0.036	RI, MS, Std
1103	1102	Nonanal	A	0.042 ± 0.003	-	RI, MS
1166	1166	δ-Terpineol	OM	0.024 ± 0.002	-	RI, MS
1176	1177	Terpinen-4-ol	OM	0.099 ± 0.002	-	RI, MS
1190	1189	α-Terpineol	OM	0.824 ± 0.027	0.738 ± 0.162	RI, MS
1228	1215	Linalool formate	OM	0.017 ± 0.001	0.017 ± 0.003	RI, MS
1308	1305	Undecanal	A	0.073 ± 0.007	-	RI, MS
1340	1341	δ-EIemene	SH	0.028 ± 0.001	-	RI, MS
1352	1351	α-Cubebene	SH	0.478 ± 0.000	0.349 ± 0.191	RI, MS
1368	1368	Cyclosativene	SH	0.220 ± 0.035	-	RI, MS
1381	1376	α-Copaene	SH	9.039 ± 0.137	9.221 ± 0.258	RI, MS
1386	1388	Cedrene	SH	0.039 ± 0.008	-	RI, MS
1394	1391	β-Elemene	SH	0.459 ± 0.001	-	RI, MS
1409	1398	β-Longipinene	SH	0.576 ± 0.012	0.283 ± 0.084	RI, MS
1412	1409	α-Gurjunene	SH	0.630 ± 0.020	0.102 ± 0.006	RI, MS
1426	1418	β-Caryophyllene	SH	11.130 ± 0.065	11.430 ± 0.334	RI, MS, Std.
1430	1429	α-Ionone	OS	0.179 ± 0.006	0.108 ± 0.012	RI, MS
1432	1432	β-Gurjunene	SH	0.218 ± 0.003	0.104 ± 0.004	RI, MS
1439	1442	α-Maaliene	SH	0.064 ± 0.001	-	RI, MS
1443	1439	Aromandendrene	SH	1.058 ± 0.035	0.686 ± 0.096	RI, MS
1447	1447	Selina-5,11-diene	SH	0.059 ± 0.023	-	RI, MS
1455	1455	Geranyl acetone	OS	2.144 ± 0.018	2.393 ± 0.101	RI, MS
1460	1455	α-Caryophyllene	SH	4.906 ± 0.084	4.299 ± 0.217	RI, MS, Std.
1466	1465	Alloaromadendrene	SH	1.453 ± 0.063	1.056 ± 0.038	RI, MS
1476	1477	γ-Himachalene	SH	0.143 ± 0.031	-	RI, MS
1480	1477	γ-Muurolene	SH	1.219 ± 0.033	1.028 ± 0.025	RI, MS
1488	1485	β-Ionone	OS	0.146 ± 0.032	-	RI, MS
1491	1485	β-Eudesmene	SH	0.935 ± 0.077	0.904 ± 0.031	RI, MS
1499	1489	Ledene	SH	1.361 ± 0.107	-	RI, MS
1503	1499	α-Muurolene	SH	0.743 ± 0.063	0.709 ± 0.046	RI, MS
1519	1513	γ-Cadinene	SH	0.871 ± 0.014	-	RI, MS
1522	1527	Selina-3,7(11)-diene	SH	0.121 ± 0.010	-	RI, MS
1529	1521	(*Z*)-Calamenene	SH	4.194 ± 0.072	3.298 ± 0.136	RI, MS
1544	1541	α-Cadinene	SH	0.345 ± 0.092	-	RI, MS
1556	1562	Cadala-1(10),3,8-triene	SH	0.441 ± 0.054	-	RI, MS
1568	1564	Epiglobulol	OS	0.963 ± 0.053	1.143 ± 0.639	RI, MS
1574	1574	Palustrol	OS	0.304 ± 0.076	-	RI, MS
1577	1574	Ylangenol	OS	0.770 ± 0.293	-	RI, MS
1590	1576	Spathulenol	OS	4.244 ± 0.151	3.267 ± 0.112	RI, MS
1595	1581	Caryophyllene oxide	OS	15.751 ± 0.064	16.018 ± 0.913	RI, MS
1597	1604	2a,3,4a,7a-Tetramethyl-2,2a,4a,5,6,7,7a,7b-octahydro-1H-cyclopenta[cd]inden-7-ol	OS	1.753 ± 0.037	-	RI, MS
1601	1590	Viridiflorol	OS	1.314 ± 0.081	-	RI, MS
1613	1611	Tetradecanal	A	1.784 ± 0.053	1.610 ± 0.249	RI, MS
1620	1606	Humulene epoxide 2	OS	4.240 ± 0.010	5.144 ± 0.347	RI, MS
1628	1616	10-epi-β-Eudesmol	OS	0.139 ± 0.006	0.113 ± 0.061	RI, MS
1636	1627	Epicubenol	OS	0.826 ± 0.005	0.916 ± 0.350	RI, MS
1639	NA	Longifolenaldehyde	OS	2.432 ± 0.223	0.924 ± 0.143	RI, MS
1642	1631	Caryophylla-4(12),8(13)-dien-5α-ol	OS	2.458 ± 0.071	-	RI, MS
1650	1640	α-epi-Cadinol	OS	1.191 ± 0.052	0.329 ± 0.124	RI, MS
1653	1645	δ-Cadinol	OS	0.279 ± 0.018	0.173 ± 0.050	RI, MS
1676	1653	10-Hydroxycalamenene	OS	0.376 ± 0.045	0.160 ± 0.047	RI, MS
1683	1674	Cadalene	SH	0.456 ± 0.044	0.212 ± 0.003	RI, MS
1687	1676	Mustakone	OS	0.441 ± 0.076	-	RI, MS
1727	1729	Murolan-3,9(11)-diene-10-peroxy	SH	1.305 ± 0.109	2.079 ± 0.448	RI, MS
1777	1772	Pentadecan-1-ol	O	0.617 ± 0.014	-	RI, MS
1817	1817	Hexadecanal	A	1.763 ± 0.033	2.249 ± 0.121	RI, MS
1844	1845	Hexahydrofarnesyl acetone	OS	0.373 ± 0.003	-	RI, MS
1892	1903	Homosalate	E	0.088 ± 0.012	-	RI, MS
1901	1906	Heptadecan-2-one	K	0.227 ± 0.016	-	RI, MS
1923	1922	Farnesyl acetone	OS	4.316 ± 0.072	4.879 ± 0.097	RI, MS
2112	2111	Phytol	OS	0.189 ± 0.006	0.337 ± 0.002	RI, MS
-	1390	6-Ethyl-2-methyldecane	O	-	0.022 ± 0.001	-
	1532	Cyperene	SH	-	0.305 ± 0.022	-
-	1589	Isocaryophyllene	SH	-	0.136 ± 0.004	-
-	1629	Rotundene	SH	-	0.091 ± 0.021	-
-	1698	Viridiflorene	SH	-	0.514 ± 0.029	-
-	1725	α-Selinene	SH	-	0.487 ± 0.029	-
-	1718	Heptadec-8-ene	SH	-	0.095 ± 0.007	-
-	1742	δ-Cadinene	SH	-	2.154 ± 0.089	-
-	1814	Tridecan-2-one	K	-	0.017 ± 0.002	-
-	1915	γ-Dehydro-ar-himachalene	SH	-	0.140 ± 0.003	-
-	1921	α-Calacorene	SH	-	0.168 ± 0.051	-
-	NA	5,5-Dimethyl-4-[(1E)-3-methyl-1,3-butadienyl]-1-oxaspiro[2.5]octane	O	-	0.197 ± 0.052	-
-	2043	Ledol	OS	-	0.142 ± 0.013	-
-	2063	Cubenol	OS	-	0.917 ± 0.201	-
-	2175	τ-Cadinol	OS	-	0.219 ± 0.010	-
-	NA	3β,9β-Dihydroxy-3,5α,8-trimethyl tricyclo[6.3.1.0(1,5)] dodecane	O	-	2.080 ± 0.102	-
-	NA	Diepicedrene-1-oxide	OS	-	1.164 ± 0.029	-
-	NA	Undec-10-ynoic acid, tetradecyl ester	E	-	0.841 ± 0.008	-
-	NA	11,11-Dimethyl-4,8-dimethylenebicyclo[7.2.0]undecan-3-ol	OS	-	1.499 ± 0.054	-
-	NA	Germacra-4(15),5,10(14)-trien-1β-ol	OS	-	1.890 ± 0.061	-
-	NA	Retinal	D	-	0.119 ± 0.021	-
		Total identified [%]		93.775	90.128	

^a^ RI = retention index. Obs. = retention index determined relative to a homologous series of *n*-alkanes (C_8_–C_40_) using a HP-5MS column. Lit. = literature RI value [10,11]. ^b^ Cl. = chemical classification; A—aldehyde, DH—diterpene hydrocarbon, E—ester, K—ketone, MH—monoterpene hydrocarbon, O—other, OD—oxygenated diterpene, OM—oxygenated monoterpene, OS—oxygenated sesquiterpene, SH—sesquiterpene hydrocarbon, ^c^ relative peak area percentage as the mean of three measurements. ^d^ Identification method: MS = mass spectrum was identical to that of National Institute of Standards and Technology Library (ver. 2.0.f), RI = the retention index matching literature database; Std = constituent identity confirmed by co-injection of authentic standards. ^e^ Identification on DB-WAX was confirmed based on the MS spectrum. NA = RI values not available in the literature.

**Table 2 plants-13-03555-t002:** *Piper philippinum* aerial part essential oil chemical composition.

^a^ RI		Compounds	^b^ Cl.	^c^ Content [%]		^d,e^ Identification
Obs.	Lit.			HP-5MS	DB-WAX	HP-5MS
929	939	α-Pinene	MH	0.068 ± 0.003	0.044 ± 0.002	RI, MS, Std
944	953	Camphene	MH	0.352 ± 0.013	0.256 ± 0.029	RI, MS, Std
1026	1031	Limonene	MH	0.048 ± 0.003	0.047 ± 0.005	RI, MS
1028	1033	Eucalyptol	OM	0.085 ± 0.014	0.057 ± 0.003	RI, MS
1099	1098	Linalool	OM	0.467 ± 0.026	0.518 ± 0.018	RI, MS, Std
1197	1195	Estragole	OM	1.232 ± 0.075	0.973 ± 0.009	RI, MS
1340	1339	δ-EIemene	SH	0.053 ± 0.005	-	RI, MS
1352	1351	α-Cubebene	SH	0.214 ± 0.009	0.131 ± 0.006	RI, MS
1374	1373	Eugenol	OM	4.662 ± 0.061	8.462 ± 0.117	RI, MS
1378	1376	α-Copaene	SH	1.979 ± 0.298	0.945 ± 0.028	RI, MS
1387	1384	β-Bourbonene	SH	0.273 ± 0.041	0.189 ± 0.024	RI, MS
1393	1391	β-Elemene	SH	1.079 ± 0.049	-	RI, MS
1405	1401	Methyleugenol	OM	0.331 ± 0.027	0.449 ± 0.032	RI, MS
1417	1415	(*Z*)-α-Bergamotene	SH	0.039 ± 0.007	-	RI, MS
1423	1418	β-Caryophyllene	SH	3.486 ± 0.185	4.367 ± 0.300	RI, MS, Std
1432	1432	β-Copaene	SH	0.364 ± 0.004	0.196 ± 0.018	RI, MS
1441	1440	Aromadendrene	SH	0.280 ± 0.008	-	RI, MS
1447	1447	Selina-5,11-diene	SH	0.609 ± 0.001	0.490 ± 0.047	RI, MS
1459	1454	Humulene	SH	1.649 ± 0.093	1.527 ± 0.034	RI, MS
1471	1467	Ishwarane	SH	25.937 ± 0.801	25.280 ± 0.450	RI, MS
1481	1477	γ-Muurolene	SH	4.591 ± 0.198	4.807 ± 0.079	RI, MS
1485	1480	Germacrene D	SH	0.564 ± 0.188	0.249 ± 0.224	RI, MS
1489	1487	Aristolochene	SH	1.604 ± 0.193	1.137 ± 0.011	RI, MS
1491	1485	β-Eudesmene	SH	1.400 ± 0.190	1.965 ± 0.033	RI, MS
1498	1491	Valencene	SH	3.363 ± 0.748	-	RI, MS
1499	1494	α-Selinene	SH	1.964 ± 0.084	3.414 ± 0.076	RI, MS
1503	1499	α-Muurolene	SH	0.452 ± 0.065	0.071 ± 0.015	RI, MS
1510	1503	β-Bisabolene	SH	0.094 ± 0.010	-	RI, MS
1518	1513	γ-Cadinene	SH	0.732 ± 0.018	-	RI, MS
1523	1522	α-Maaliene	SH	0.479 ± 0.017	-	RI, MS
1527	1524	δ-Cadinene	SH	2.614 ± 0.117	2.659 ± 0.171	RI, MS
1537	1535	Cubenene	SH	0.136 ± 0.014	0.107 ± 0.003	RI, MS
1542	1538	α-Cadinene	SH	0.134 ± 0.006	-	RI, MS
1548	1546	α-Calacorene	SH	0.226 ± 0.014	0.142 ± 0.022	RI, MS
1568	1565	Nerolidol	OS	9.372 ± 0.680	10.519 ± 0.195	RI, MS
1584	1576	Spathulenol	OS	0.461 ± 0.030	0.473 ± 0.008	RI, MS
1590	1581	Caryophyllene oxide	OS	0.956 ± 0.044	0.722 ± 0.074	RI, MS
1617	1606	Humulene epoxide 2	OS	0.365 ± 0.014	0.296 ± 0.026	RI, MS
1621	1630	α-Acorenol	OS	0.146 ± 0.006	-	RI, MS
1635	1642	Cubenol	OS	0.462 ± 0.009	0.238 ± 0.081	RI, MS
1644	1644	10,10-Dimethyl-2,6-dimethylenebicyclo[7.2.0]undecan-5-ol	OS	0.114 ± 0.012	-	RI, MS
1649	1640	α-epi-Muurolol	OS	0.325 ± 0.055	-	RI, MS
1653	1645	δ-Cadinol	OS	0.259 ± 0.050	0.110 ± 0.021	RI, MS
1663	1662	Neointermedeol	OS	0.815 ± 0.084	-	RI, MS
1668	1669	Intermedeol	OS	0.565 ± 0.058	0.646 ± 0.019	RI, MS
1682	1685	Eudesma-4(15),7-dien-1β -ol	OS	1.141 ± 0.092	1.516 ± 0.082	RI, MS
1690	1680	Germacra-4(15),5,10(14)-trien-1α-ol	OS	0.518 ± 0.009	0.318 ± 0.054	RI, MS
1691	1680	3-Ishwarone	OS	6.916 ± 0.142	2.588 ± 0.141	RI, MS
1769	1763	Aristolone	OS	0.127 ± 0.009	-	RI, MS
1776	1778	β-Cosol	OS	2.677 ± 0.060	2.945 ± 0.104	RI, MS
1804	1805	τ-Cadinol acetate	E/OS	0.128 ± 0.011	0.119 ± 0.003	RI, MS
1814	1831	Valerenyl acetate	E/OS	0.720 ± 0.043	0.433 ± 0.004	RI, MS
2112	2111	Phytol	OD	0.731 ± 0.069	1.067 ± 0.015	RI, MS
2217	2218	Phytol, acetate	E/OD	0.164 ± 0.010	-	RI, MS
-	1488	Ylangene	SH	-	0.197 ± 0.001	-
-	1603	α-Guaiene	SH	-	0.163 ± 0.034	-
-	1832	Cadina-1,3,5-triene	SH	-	0.549 ± 0.037	-
-	1895	Epicubebol	OS	-	0.106 ± 0.004	-
-	1924	Tetradecanal	A	-	0.072 ± 0.005	-
-	NA	β-Cyperone	OS	-	3.352 ± 0.057	-
-	2104	Globulol	OS	-	0.635 ± 0.057	-
-	2299	Aromadendrenepoxide	OS	-	0.112 ± 0.015	-
-	2910	Hexadecanoic acid	FA	-	3.391 ± 0.063	-
		Total identified [%]		88.524	89.045	

^a^ RI = retention index. Obs. = retention index determined relative to a homologous series of *n*-alkanes (C_8_–C_40_) using a HP-5MS column. Lit. = literature RI value [10,11]. ^b^ Cl. = chemical classification; A—aldehyde, DH—diterpene hydrocarbon, E—ester, K—ketone, MH—monoterpene hydrocarbon, O—other, OD—oxygenated diterpene, OM—oxygenated monoterpene, OS—oxygenated sesquiterpene, SH—sesquiterpene hydrocarbon, ^c^ relative peak area percentage as the mean of three measurements. ^d^ Identification method: MS = mass spectrum was identical to that of National Institute of Standards and Technology Library (ver. 2.0.f), RI = the retention index matching literature database; Std = constituent identity confirmed by co-injection of authentic standards. ^e^ Identification on DB-WAX was confirmed based on the MS spectrum. NA = RI values not available in the literature.

## Data Availability

The original contributions presented in the study are included in the article; further inquiries can be directed to the corresponding author.

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
