# Peer review of "Evaluation of Chemical Composition and Anti-Staphylococcal Activity of Essential Oils from Leaves of Two Indigenous Plant Species, Litsea leytensis and Piper philippinum"

_plants, 2024, doi:10.3390/plants13243555_

Round 1

Reviewer 1 Report

Comments and Suggestions for Authors

I have made my comments in the attached pdf file.

Kind regards.

Author Response

Query 1: typo- put in Italic.

Response: typo corrected

Query 2: Can authors provide on what basis you assess that this MIC is not significant? This is quite low MIC since it is in microgram/mL.

Response: With aim to provide basis for MIC assessment, we added following text: According to the recommendations previously proposed for more effective assessment of anti-infective potential of natural products, minimum inhibitory concentrations (MICs) below 100 μg/ml for EOs should be considered as promising activity. In addition, samples with MICs higher than 1000 μg/ml should strictly be evaluated as no active and excluded from further experiments [9]. Based on these criteria,……..

Following reference has been added to their list:

Kokoska, L.; Kloucek, P.; Leuner, O.; Novy, P. Plant-derived products as antibacterial and antifungal agents in human health care. Curr. Med. Chem. 2019, 26, 1–38.

Query 3: I think it should be "are" since you are referring to the "results" in plural? Check/correct.

Response: grammar corrected.

Query 4: In English content is uncountable noun and should not be used in plural in this context.

Response: grammar corrected.

Query 5: And why these compounds did not exhibit biological activity in antifungal assay applied here? Please suggest some explanation. Will selection of some other microbes maybe will provide different results?

Response: With aim to provide explanation for lack of antibacterial effect of EOs tested in the study, we added following text: In previous experiments investigating their antimicrobial properties, caryophyllene oxide, β-caryophyllene and α-copaene produced moderate to weak in vitro growth-inhibitory effect against Staphylococcus aureus with MICs 125, 250 and ~1000 μg/ml, respectively [19,20]. The relatively high MIC values previously detected for all three compounds can explain absence of anti-staphylococcal activity of EOs assayed in this study.

Following references have been added to the list:

Ben Hsouna, A.; Ben Halima, N.; Abdelkafi, S.; Hamdi, N. Essential oil from Artemisia phaeolepis: Chemical composition and antimicrobial activities. J. Oleo Sci. 2013, 62, 973-980.

Chen, S.Y.; Zheng, H.; Yang, S.P.; Qi, Y.G.; Li, W.; Kang, S.N.; Hu, H.; Hua, Q.; Wu, Y.K.; Liu, Z.J. Antimicrobial activity and mechanism of α-copaene against foodborne pathogenic bacteria and its application in beef soup. LWT - Food Sci. Technol. 2024, 195, 115848.

Please provide pictures of plants examined. It would be quite informative for future readers.

Unfortunately, we do not have high quality photographs showing typical botanical features of the species analysed available now.

Query 4: was the predominant class"- this is one class with several compounds consisting it.

Response: The sentence's grammar was changed to singular.

Reviewer 2 Report

Comments and Suggestions for Authors

Review of the article: “Evaluation of chemical composition and anti-staphylococcal activity of essential oils from leaves of two indigenous plant species, Litsea leytensis and Piper philippinum, from Leyte, Philippines”

The article addresses an interesting topic regarding the analysis of the chemical composition of essential oils derived from two indigenous Philippine plant species: Litsea leytensis and Piper philippinum. The authors provide a detailed overview of the significance of these plant species and introduce the methods used for essential oil analysis, such as gas chromatography and mass spectrometry. The introduction provides a solid research background, including the botanical context and potential biological applications of essential oils from these species, which reinforces the article’s relevance.

This study is innovative as it offers information on the chemical composition and antibacterial activity of essential oils from two plant species that have not been previously studied in detail. The article makes a significant contribution to the literature, particularly regarding the potential applications of essential oils in the pharmaceutical and cosmetic industries. The study's importance is high, as exploring new sources of phytochemicals with potential biological properties can support further discoveries in the field of natural antimicrobial agents.

The article is well-organized, and the results are clearly presented, especially in tables and figures illustrating the chemical composition of the primary components of the essential oils. The language is correct, though minor edits could improve the text’s fluency and precision in scientific terminology.

The study was conducted rigorously, and the description of the methods, including chromatographic techniques and tests for the presence of Staphylococcus aureus, is detailed. The authors explain the use of different polarity columns and the microdilution method to assess antibacterial activity. Although the essential oils did not show antibacterial activity against S. aureus, the detailed chemical composition analysis provides a foundation for further research into other biological properties of the studied species.

The main assumption of the study focused on evaluating the antibacterial activity of the oils against Staphylococcus aureus; however, the results showed no efficacy against this strain. If the oils do not exhibit promising antibacterial properties, it would be beneficial to consider expanding the analysis to other bacteria or pathogens that may be more susceptible. Alternatively, the article could suggest more detailed studies on other biological properties of these oils.

While the authors identified several significant chemical compounds in the oils, known for their biological activities (e.g., β-caryophyllene and nerolidol), a more in-depth discussion of the potential applications of these compounds would enhance the significance of the findings.

The chemical composition of essential oils can vary depending on environmental factors (e.g., soil, climate, altitude). It would be beneficial to mention this context and consider further research on the impact of environmental conditions on chemical composition.

Only hydrodistillation was used to obtain the oils in this study. Other methods, such as steam distillation or COâ‚‚ extraction, could yield different chemical profiles. Exploring various extraction methods could complement the results and provide additional information on the composition of the oils.

Although the language of the article is clear, some minor adjustments could improve the text’s coherence and make the results easier to interpret.

The conclusions suggest further studies but could be expanded with specific directions, e.g., research on antifungal, antioxidant, or anticancer activities.

Comments and Questions for the Authors:

  1. Could the authors discuss how environmental differences in the habitats of Litsea leytensis and Piper philippinum might affect the chemical composition of the oils? This could enrich the conclusions regarding phytochemical variability.
  2. Since the essential oils did not show antibacterial activity against S. aureus, do the authors consider testing the same oils on other bacterial strains that might be more susceptible to these substances?
  3. Do the authors plan to investigate other potential biological properties, such as antioxidant, antifungal, or anticancer activities, which could have applications in the cosmetic or pharmaceutical industries?
  4. Hydrodistillation is an effective method, but it may affect the chemical composition of essential oils. Did the authors consider using other extraction methods, such as steam distillation or COâ‚‚ extraction, to examine differences in chemical composition?
  5. Can the authors elaborate on the biological significance of the main compounds detected in the oils? For instance, β-caryophyllene and nerolidol are known for various biological effects. Expanding on this aspect could help readers better understand the potential applications of these oils.

The article is a valuable contribution to research on essential oils from endemic plant species, providing data on the chemical composition and preliminary assessment of their antimicrobial potential. Although no antibacterial activity was observed, this study opens avenues for further exploration of the biological potential of essential oils from Litsea leytensis and Piper philippinum.

Author Response

Query 1: Could the authors discuss how environmental differences in the habitats of Litsea leytensis and Piper philippinum might affect the chemical composition of the oils? This could enrich the conclusions regarding phytochemical variability.

Response: With aim to discuss influence of environmental factors on chemical composition of studied essential oils (EOs), we added following sentences to the end of second paragraph of the Results and Discussion section: Reported chemical variability of EOs obtained from leaves of L. cubeba of different geographical origin [24] suggests that environmental factors can also affect the phytochemical variability of other species of the genus, including composition of L. leytensis EO.

We also added following sentence to the end of third paragraph of the Results and Discussion section: Since considerable variations in chemical composition of EOs have previously been demonstrated for different populations of P. nigrum grown in China [41], the potential influence of environmental conditions on chemistry of plants belonging to Piper genus should be considered prior the industrial utilization of their EOs.

Following references have been added to their list:

Bighelli, A.; Muselli, A.; Casanova, J.; Tam, N.T.; Van Anh, V.; Bessière, J.M. Chemical variability of Litsea cubeba leaf oil from Vietnam. J. Essent. Oil Res. 2005, 17, 86-88.

Li, Y.X.; Zhang, C.; Pan, S.Y.; Chen, L.; Liu, M.; Yang, K.L.; Zeng, X.B.; Tian, J. Analysis of chemical components and biological activities of essential oils from black and white pepper (Piper nigrum L.) in five provinces of southern China. LWT - Food Sci. Technol. 2020, 117, 108644.

Query 2 and 3: Since the essential oils did not show antibacterial activity against S. aureus, do the authors consider testing the same oils on other bacterial strains that might be more susceptible to these substances? Do the authors plan to investigate other potential biological properties, such as antioxidant, antifungal, or anticancer activities, which could have applications in the cosmetic or pharmaceutical industries?

Response: With aim to provide recommendation for the future research directions, we modified last sentence of the Conclusions section as follows: However, the presence of bioactive compounds in the EOs suggests their potential for future investigation, which should focus on the determination of growth-inhibitory activity against other bacterial strains as well as on evaluation of other biological properties, such as anticancer, antifungal and antioxidant effects, which could have applications in the cosmetic and pharmaceutical industries.

Query 4: Hydrodistillation is an effective method, but it may affect the chemical composition of essential oils. Did the authors consider using other extraction methods, such as steam distillation or COâ‚‚ extraction, to examine differences in chemical composition?

Response: With aim to discuss potential influence of extraction method on chemical composition of essential oils studied, we added following sentences to the end of second paragraph of the Results and Discussion section: In addition, the experiment comparing solvent-free microwave extraction and hydrodistillation showed that the chemical composition of L. cubeba EO could be influenced by both the extraction method used [25], which should be considered for further scaling-up of extraction of volatile compounds from species of Litsea genus.

We also added following sentence to the end of third paragraph of the Results and Discussion section: It has also recently been reported that the chemical composition of P. nigrum hydrodistlied EO varies form its supercritical carbon dioxide extract [42]. This result suggests that the analysis of composition of volatile compounds present in the Piper genus can be influenced by extraction method used.

Following references have been added to their list:

Qiu, Y.F.; Wang, Y.; Li, Y. Solvent-free microwave extraction of essential oils from Litsea cubeba (Lour.) Pers. at different harvesting times and their skin-whitening cosmetic potential. Antioxidants 2022, 11, 2389.

Vihanova, K.; Urbanova, K.; Nguon, S.; Kokoska, L. Chemical composition of essential oils and supercritical carbon dioxide extracts from Amomum kravanh, Citrus hystrix and Piper nigrum ‘Kampot’. Molecules 2023, 28, 7748.

Query 5: Can the authors elaborate on the biological significance of the main compounds detected in the oils? For instance, β-caryophyllene and nerolidol are known for various biological effects. Expanding on this aspect could help readers better understand the potential applications of these oils.

With aim to discuss more in details biological significance of the main compounds detected in essential oils, we modified sixth sentence of the second paragraph of the Results and Discussion section as follows: For example, analgesic, antibacterial, anticancer, antifungal, anti-inflammatory, antioxidant, cardioprotective, gastroprotective, hepatoprotective, immunomodulatory, nephroprotective, and neuroprotective effects were reported for β-caryophyllene [12-14].

We also modified fourth sentence of third paragraph of the Results and Discussion section: Nerolidol is an economically important sesquiterpene since it is predominantly utilised as a fragrance component in the perfume industry [24] with known anticancer, anti-inflammatory, antimicrobial, antinociceptive, antioxidant, and antiparasitic (anti-leishmanial, -malarial, -schistosomal and -trypanosomal) activities [31-34].

Following references have been added to their list:

Machado, K.D.; Islam, M.T.; Ali, E.S.; Rouf, R.; Uddin, S.J.; Dev, S.; Shilpi, J.A.; Shill, M.C.; Reza, H.M.; Das, A.K.; Shaw, S.; Mubarak, M.S.; Mishra, S.K.; Melo-Cavalcante, A.A.D. A systematic review on the neuroprotective perspectives of beta-caryophyllene. Phytother. Res. 2018, 32, 2376-2388.

Chan, W.K.; Tan, L.T.H.; Chan, K.G.; Lee, L.H.; Goh, B.H. Nerolidol: A sesquiterpene alcohol with multi-faceted pharmacological and biological activities, Molecules 2016, 21, 529.

Reviewer 3 Report

Comments and Suggestions for Authors

This manuscript evaluated the chemical composition and anti-staphylococcal activity of essential oils from leaves of two indigenous plant species Litsea leytensis and Piper philippinum from Leyte, Philippines. Their approach is interesting but it has flaws that make this version unacceptable for publication. While I have no objection against publishing the data, I have some issues that need addressing:

1. Statistical methods in results are needed. Information about statistical methods obtained in the experiments must be provided and detailed in methods section.

2. In results, chemical analysis of plant essential oil needs to be checked in triplicate.

3. I am concerned about the poorly elements in the discussion; authors must be provided a good debate for results interpretation.

A few points:

Ls.4-5: Delete “from Leyte, Philippines”.

Ls.37-74: More information about this plant species is needed (botanical, ecological, and industrial use).

L.60: Also, a hypothesis for this study is needed.

L.142: In results section, information about antimicrobial assay is missing.

Ls.224-225: Quantitative analysis should be clearly explained in this section.

L.225: There not statistical analysis.

L.227: Delete “In summary,”.

Author Response

Query 1. Statistical methods in results are needed. Information about statistical methods obtained in the experiments must be provided and detailed in methods section.

Response: With aim to provide information on statistical evaluation of data obtained, we added following sentence to the end of the GC-MS Analysis section: The analysis of each EO was performed in three independent measurements and the relative peak area percentage was expressed as an average plus/minus standard deviation.

Query 2. In results, chemical analysis of plant essential oil needs to be checked in triplicate.

Response: We provided averages with standard deviations for percentage content of compounds identified in the essential oils analysed in the Tables 1 and 2.

Query 3. I am concerned about the poorly elements in the discussion; authors must be provided a good debate for results interpretation.

Response: With aim to discuss obtained results more in detail we added/modified text of the discussion as follows:

With aim to discuss influence of environmental factors on chemical composition of studied essential oils (EOs), we added following sentences to the end of second paragraph of the Results and Discussion section: Reported chemical variability of EOs obtained from leaves of L. cubeba of different geographical origin [24] suggests that environmental factors can also affect the phytochemical variability of other species of the genus, including composition of L. leytensis EO.

We also added following sentence to the end of third paragraph of the Results and Discussion section: Since considerable variations in chemical composition of EOs have previously been demonstrated for different populations of P. nigrum grown in China [41], the potential influence of environmental conditions on chemistry of plants belonging to Piper genus should be considered prior the industrial utilization of their EOs.

Following references have been added to their list:

Bighelli, A.; Muselli, A.; Casanova, J.; Tam, N.T.; Van Anh, V.; Bessière, J.M. Chemical variability of Litsea cubeba leaf oil from Vietnam. J. Essent. Oil Res. 2005, 17, 86-88.

Li, Y.X.; Zhang, C.; Pan, S.Y.; Chen, L.; Liu, M.; Yang, K.L.; Zeng, X.B.; Tian, J. Analysis of chemical components and biological activities of essential oils from black and white pepper (Piper nigrum L.) in five provinces of southern China. LWT - Food Sci. Technol. 2020, 117, 108644.

With aim to discuss potential influence of extraction method on chemical composition of essential oils studied, we added following sentences to the end of second paragraph of the Results and Discussion section: In addition, the experiment comparing solvent-free microwave extraction and hydrodistillation showed that the chemical composition of L. cubeba EO could be influenced by both the extraction method used [25], which should be considered for further scaling-up of extraction of volatile compounds from species of Litsea genus.

We also added following sentence to the end of third paragraph of the Results and Discussion section: It has also recently been reported that the chemical composition of P. nigrum hydrodistlied EO varies form its supercritical carbon dioxide extract [42]. This result suggests that the analysis of composition of volatile compounds present in the Piper genus can be influenced by extraction method used.

Following references have been added to their list:

Qiu, Y.F.; Wang, Y.; Li, Y. Solvent-free microwave extraction of essential oils from Litsea cubeba (Lour.) Pers. at different harvesting times and their skin-whitening cosmetic potential. Antioxidants 2022, 11, 2389.

Vihanova, K.; Urbanova, K.; Nguon, S.; Kokoska, L. Chemical composition of essential oils and supercritical carbon dioxide extracts from Amomum kravanh, Citrus hystrix and Piper nigrum ‘Kampot’. Molecules 2023, 28, 7748.

Query 4. Ls.4-5: Delete “from Leyte, Philippines”.

Response: deleted

Query 5. Ls.37-74: More information about this plant species is needed (botanical, ecological, and industrial use).

Response: With aim to provide more information on the species we added/modified text of the introduction as follows: Another example of a native Philippine plant is Piper philippinum Miq. (Piperaceae), which is a woody climber distributed throughout the Southern China, Taiwan, and Philippines in thickets and forests at low and medium altitudes [7]. The older stems are traditionally used for betel quid chewing, together with lime and catechu.

Query 6. L.60: Also, a hypothesis for this study is needed.

Response: With aim to provide hypothesis for the study, we added/modified text of the second last sentence of the Introduction section as follows: Since the certain species of the genera Litsea and Piper (e.g. Litsea cubeba and Piper nigrum) are important industrial sources of EOs, we decided to determine the chemical composition of the volatile compounds present in the leaves of two indigenous Philippine plants, namely L. leytensis and P. philippinum, using gas chromatography and mass spectrometry (GC-MS).

Query 7. L.142: In results section, information about antimicrobial assay is missing.

Response: Since the antimicrobial susceptibility testing of both EOs did not show any antimicrobial effect at the range of concentrations tested, we provided just brief description of the results in the first paragraph of the Results and Discussion section. We added/modified the text as follows: According to the recommendations previously proposed for more effective assessment of anti-infective potential of natural products, minimum inhibitory concentrations (MICs) below 100 μg/ml for EOs should be considered as promising activity. In addition, samples with MICs higher than 1000 μg/ml should strictly be evaluated as no active and excluded from further experiments [9]. Based on these criteria, the EOs did not show any antibacterial activity against S. aureus in the liquid nor vapor phases (MICs >1,024 μg/mL).

Query 8. Ls.224-225: Quantitative analysis should be clearly explained in this section.

Response: In correspondence with international standards for antimicrobial susceptibility testing (CLSI, FDA, ISO), we used mode and median values of three independent experiments (each performed in triplicate) for MICs evaluation. We added/modified the text as follows: All experiments were performed in triplicate, in three independent experiments. According to the widely accepted norm in MIC testing, the mode and median were used for the final value calculation when the triplicate endpoints were within the two- and three-dilution range, respectively [47].

Frankova, A.; Vištejnova, L.; Merinas-Amo, T.; Leheckova, Z.; Doskocil, I.; Wong Soon, J.; Kudera, T.; Laupua, F.; Alonso-Moraga, A.; Kokoska, L. In vitro antibacterial activity of extracts from Samoan medicinal plants and their effect on proliferation and migration of human fibroblasts. J. Ethnopharmacol. 2021, 264, 113220.

Query 9. L.225: There not statistical analysis.

We provided averages with standard deviations for percentage content of compounds identified in the EOs analysed in the Tables 1 and 2. Because the main aim of the research was the first qualitative analysis of EOs of two taxonomically distinct species, which were isolated and analysed using the same methods, we do not provide other statistical evaluation, as it is common for similar studies.

Query 10. L.227: Delete “In summary,”.

Response: expression deleted.

Round 2

Reviewer 2 Report

Comments and Suggestions for Authors

The article titled "Evaluation of chemical composition and anti-staphylococcal activity of essential oils from leaves of two indigenous plant species Litsea leytensis and Piper philippinum from Leyte, Philippines" presents the results of research on essential oils derived from two endemic plant species in the Philippines. The authors analyzed their chemical composition using GC-MS and evaluated their activity against Staphylococcus aureus. The study addresses an interesting topic, focusing on underexplored endemic plants with potential industrial and scientific importance.

The greatest strength of the article lies in its scientific novelty, as it is the first study to describe the chemical composition and bioactivity of the essential oils from these two plant species. The research was conducted rigorously, utilizing appropriate analytical techniques such as GC-MS with columns of different polarity. The results are detailed, and the structure of the article enables the reader to understand both the methodology and the findings. Of particular value is the identification of bioactive compounds in the oils, which highlights their potential applications in the cosmetic or pharmaceutical industries, despite the lack of antibacterial activity against S. aureus.

However, despite its merits, the article requires some revisions. In particular, certain sections of the text lack clarity, especially the conclusions and abstract. The narrative in these parts should be simplified and more focused on the key findings of the study. Additionally, the article contains minor editorial errors, such as typos and inconsistencies in the naming of chemical compounds (e.g., "β-Caryophyllene"). These issues hinder smooth reading and could negatively impact the perception of the work. The conclusions section is relatively brief and does not fully reflect the significance of the findings. It would be beneficial to expand this section to include suggestions for further research, particularly on other potential biological properties of the oils, such as antioxidant or antifungal effects.

The corrections introduced so far have partially improved the text, especially in the presentation of tables and references to the literature. The chemical analysis data have been organized, and the scientific context has been enriched. However, the revisions have not fully addressed issues with unclear narrative flow and insufficient development of the conclusions.

In summary, the article in its current form is not ready for publication. I strongly recommend implementing additional revisions, particularly in enhancing the conclusions, improving the language style, and eliminating editorial errors. Once these suggestions are addressed, the article will be a valuable contribution to the field of essential oil research and its applications.

Author Response

Query 1: However, despite its merits, the article requires some revisions. In particular, certain sections of the text lack clarity, especially the conclusions and abstract. The narrative in these parts should be simplified and more focused on the key findings of the study.

Response 1: With aim to improve clarity, we modified text of the abstract as follows:

Many indigenous plants of Philippines, including essential oil-bearing species, remain phytochemically and pharmacologically unexplored. In this study, the chemical composition of leaf essential oils (EOs) hydrodistilled from Litsea leytensis (Lauraceae) and Piper philippinum (Piperaceae) was determined using dual-column (HP-5MS/DB-WAX)/dual-detector gas chromatography and mass spectrometry analysis. Caryophyllene oxide (15.751/16.018%) was identified as main compound of L. leytensis EO, followed by β-caryophyllene (11.130/11.430%) and α-copaene (9.039/9.221%). Ishwarane (25.937/25.280%), nerolidol (9.372/10.519%) and 3-ishwarone (6.916/2.588%) were the most abundant constituents of P. philippinum EO. Additionally, the in vitro growth-inhibitory activity of the EOs in the liquid and vapor phases against Staphylococcus aureus was evaluated using the broth microdilution volatilization assay. Although the results showed no anti-staphylococcal effect, the presence of various bioactive compounds in both EOs suggests their potential future use in the industrial applications.

In addition, we modified text of the Conclusions section as follows:

This is the first report of the chemical composition of EOs hydrodistilled from the leaves of L. leytensis (Lauraceae) and P. philippinum (Piperaceae), which provides new insides into the phytochemistry of these indigenous plant species found in the Philippines. Sesquiterpenoids, namely caryophyllene oxide, α-copaene and β-caryophyllene (L. leytensis) and 3-ishwarone, ishwarane, and nerolidol (P. philippinum), was the predominant class of compounds identified in both EOs. Using GC-MS equipped with two columns of differing polarity, isomeric cadinenes were detected in L. leytensis EO and copaenes and ylangene were identified in P. philippinum EO. Although the assessment of the antibacterial activity of the EOs showed no growth-inhibitory effect on S. aureus, the presence of bioactive compounds such as caryophyllene oxide, β-caryophyllene, α-copaene, ishwarane and nerolidol in the EOs suggests their potential future use in the cosmetic and pharmaceutical applications. However, the evaluation of safety and broader spectrum of biological properties, such as anticancer, antifungal and antioxidant effects, will be necessary before their possible industrial usage.

Query 2: Additionally, the article contains minor editorial errors………

Response 2: We also corrected several typos and inconsistencies in the naming of chemical compounds.

Reviewer 3 Report

Comments and Suggestions for Authors

The authors have incorporated all suggestions and reviewer comments into the latest version, now the manuscript seems much clear. There are minor points to be corrected:

Ls.74, 75 and 125: Change ml by mL.

Ls.124-125: "in vitro" should be in italic.

L.164: Correct "hydrodistlied".

Author Response

Query 1: Ls.74, 75 and 125: Change ml by mL.

Response 1: Corrected.

Query 2: Ls.124-125: "in vitro" should be in italic.

Response 2: Corrected.

Query 3: L.164: Correct "hydrodistlied".

Response 3: Corrected.